# Spatio-Temporal Dynamics of Diffusion-Associated Deformations of Biological Tissues and Polyacrylamide Gels Observed with Optical Coherence Elastography

**DOI:** 10.3390/ma16052036

**Published:** 2023-03-01

**Authors:** Yulia M. Alexandrovskaya, Ekaterina M. Kasianenko, Alexander A. Sovetsky, Alexander L. Matveyev, Vladimir Y. Zaitsev

**Affiliations:** 1Institute of Applied Physics of the Russian Academy of Sciences, Uljanova St., 46, 603950 Nizhny Novgorod, Russia; 2Federal Scientific Research Center “Crystallography and Photonics”, Institute of Photon Technologies, Russian Academy of Sciences, 2 Pionerskaya Street, Troitsk, 108840 Moscow, Russia

**Keywords:** optical coherence elastography, optical clearing, cartilage, diffusion, polyacrylamide, osmosis, osmotic strain, deformation

## Abstract

In this work, we use the method of optical coherence elastography (OCE) to enable quantitative, spatially resolved visualization of diffusion-associated deformations in the areas of maximum concentration gradients during diffusion of hyperosmotic substances in cartilaginous tissue and polyacrylamide gels. At high concentration gradients, alternating sign, near-surface deformations in porous moisture-saturated materials are observed in the first minutes of diffusion. For cartilage, the kinetics of osmotic deformations visualized by OCE, as well as the optical transmittance variations caused by the diffusion, were comparatively analyzed for several substances that are often used as optical clearing agents, i.e., glycerol, polypropylene, PEG-400 and iohexol, for which the effective diffusion coefficients were found to be 7.4 ± 1.8, 5.0 ± 0.8, 4.4 ± 0.8 and 4.6 ± 0.9 × 10^−6^ cm^2^/s, respectively. For the osmotically induced shrinkage amplitude, the influence of the organic alcohol concentration appears to be more significant than the influence of its molecular weight. The rate and amplitude of osmotically induced shrinkage and dilatation in polyacrylamide gels is found to clearly depend on the degree of their crosslinking. The obtained results show that observation of osmotic strains with the developed OCE technique can be applied for structural characterization of a wide range of porous materials, including biopolymers. In addition, it may be promising for revealing alterations in the diffusivity/permeability of biological tissues that are potentially associated with various diseases.

## 1. Introduction

Passive diffusion, as a method of matter exchange and establishing equilibria, occurs both in living and inanimate matter, for example, during nutrient delivery within tissues and organs, bioimplant fabrication or moisture saturation of polymers. Optical clearing of biological tissues, intended to improve the conditions of optical diagnostics, also utilizes short-term saturation of tissues with optical clearing agents [1,2]. In the presence of significant concentration gradients that are typical of such situations, diffusion processes in porous media can be accompanied by significant osmotic deformations; the nature, amplitude and duration of which can affect the properties of the whole system. Under non-equilibrium conditions, it is extremely difficult to monitor the rate and sign of deformations using standard approaches based on mechanical measurements. In recent decades, deformations of porous materials associated with the diffusion of osmotically active solutions and/or dehydration have been considered mainly as by-pass phenomena accompanying the intended “useful” effects of such substances in biomedical procedures or industrial processes. Such osmotic deformations are common, in particular, for optical clearing of biological tissues with the application of hyperosmotic solutions [1,2,3,4,5] and for the diffusion of non-isotonic solutions through cell membranes [6,7], as well as for some technological processes of food preparation, especially drying of vegetables and fruits [8,9,10]. The attention on osmotically induced alterations of mechanical properties of the analyzed objects, including the quantification of the observed deformations, has been growing during the last few years [11,12,13,14]. Regarding biological tissues, the diagnostic prospects of diffusion measurements have attracted increasing interest, for example, for diagnosing diabetes mellitus [15,16]. It should be noted that techniques allowing for minimally invasive and even non-contact measurements of diffusion-associated deformations represent a special interest for this area of research.

Until recently, studies of osmotically induced deformations have been rather challenging because of the lack of suitable and practically convenient methods, as well as an insufficient understanding of the useful information that can be provided by observations of such deformations. In particular, in the approach which uses osmotically active substances for optical clearing of biological tissues, diffusion-induced deformations were reasonably considered as a factor violating the safety of the diagnostic procedures that should be reduced [17,18,19].

This article introduces a non-contact study of the spatially resolved dynamics of osmotically induced deformations in biological tissue and acrylamide polymeric gels, which are a good model of a porous material with composition-controlled properties, using optical coherent elastography (OCE). This research technique was proposed recently due to the development of appropriate methods of analysis of signals obtained by phase-sensitive optical coherence tomography [20].

The work continues the pilot studies on this topic, previously carried out by us and other scientific groups [14,21,22,23]. The present study is focused on demonstrating that osmotic deformations are characterized by complex spatio-temporal dynamics during osmotic agent diffusion into the material depth, and observation of these features can provide useful information about the structural properties of the objects under the study. Solutions of organic alcohols, such as glycerol, propylene-glycol (PG) and PEG-400, are used as the osmotically active substances, which are known to be strong osmotic agents and capable of inducing deformations of a sufficiently large amplitude [24,25,26,27,28,29,30,31,32]. For a comparison of their effect, a rather mild osmotic agent, iohexol, is also used [33,34]. All these components have been used for many years in different approaches of optical clearing of biological tissues, and their effect on biopolymers has been studied quite extensively. Therefore, we expect that the results of this work will be of interest both to the specialists in the field of optical diagnostics and to a wider range of specialists interested in diffusion-associated processes in polymers.

## 2. Materials and Methods

### 2.1. Polyacryalmide Hydrogels (Tissue Phantoms)

For tissue mimicking phantoms, polyacrylamide hydrogels were synthesized by previously developed techniques [35,36,37] using the following reagents: acrylamide (Sigma-Aldrich, St. Louis, MO, USA, ≥99%), NN′-methylenebis-acrylamide (99%), ammonium persulfate (Sigma-Aldrich, St. Louis, MO, USA, 98%) and TEMED (tetramethylethylenediamine) (Sigma-Aldrich, St. Louis, MO, USA, ≥99.5%) as an initiator of polymerization. In this work, the main variable parameter for hydrogels was their crosslinking degree, determined by the ratio of acrylamide and bis-acrylamide (Table 1). The preparation technique can be briefly described as follows: the quantities of acrylamide and bis-acrylamide powders indicated in Table 1 were mixed and dissolved in distilled water. Then, 0.0005 g of ammonium persulfate was added. After complete dissolution of the reagents, 1 drop (30 µL) of TEMED was added to start the polymerization. Immediately after the addition of TEMED, a part of the mixture was collected with a cylindrical syringe of 1 mm volume. The polymerized gels were obtained as cylinders with d = 10 mm and kept hydrated until use.

For complementary OCE measurements, cross-sectional cylindrical cuts of hydrogels with a diameter, d, of ≈10 mm and a thickness of ~2.0 mm were prepared using a scalpel.

### 2.2. Cartilaginous Tissue

Porcine costal cartilages of the 5th–8th ribs were taken from a local butcher immediately after slaughter and stored frozen at −15 °C. Thawing was performed stepwise: first, at 4 °C for at least 8 h, then at room temperature for 1 h. Prior to the measurements, all samples were equilibrated in saline solution containing 0.9% NaCl (~300 mOsm). Cross-sectional cylindrical cuts of cartilages (d ≈ 10 mm and thickness of ~2.0 mm) were prepared using a scalpel and a metal punch tool. Such prepared samples were used for OCT measurements. For spectrophotometry, the parameters of the samples were d ≈ 0.7 mm and various thicknesses (0.6–1.6 mm).

### 2.3. Spectrophotometry

Spectrophotometry was carried out on biological tissue only. The intensity of optical transmission was determined using an Ecroschim PE-5400VI Spectrophotometer (ECROSKHIM Co., Ltd., Saint-Petersburg, Russia) at 700 nm wavelength. The cartilaginous samples were put in quartz cuvettes and oriented so that the normal to the surface of the samples coincided with the axis of the optical beam in the measuring cell of the spectrophotometer. The cuvettes were filled with liquid and the dynamics of the optical transmission increase were recorded. All the chemicals used for the preparation of osmotically active solutions were of analytical grade and purchased from ChemReagent (Ufa, Bashkortostan, Russia). Omnipaque-300 (300 mg/mL iodine, 39.2% mass iohexol solution) was purchased from GE Healthcare Ireland (Cork, Ireland). Distilled water was used to dissolve the reagents. The following water-based solutions were used for the transmission measurements: glycerol (30% *v*/*v*), PG (30% *v*/*v*), PEG-400 (30% *v*/*v*) and iohexol (Omnipaque, 30% *v*/*v*). All the measurements were repeated at least 3–6 times using samples with different thicknesses ranging from 0.8 mm to 1.4 mm (and mechanically measured with an accuracy of 0.1 mm). These thicknesses were used for the estimation of the diffusion coefficients, the results of which are presented in Table 2.

### 2.4. Diffusion Kinetics

The diffusion kinetics corresponding to the optical clearing of the samples were estimated according to the analytical solution of Fick’s diffusion problem for the bilateral adsorption by a flat plate slab [38,39]. Diffusion through the ends of the slab was neglected. One-dimensional diffusion can be described by Fick’s second law:(1)$$\frac{\partial C(x,t)}{\partial t}=D\frac{\partial C^{2}(x,t)}{\partial x^{2}}$$
where *C* is the concentration of a diffusant, *t* is the time of diffusion, *x* is the coordinate inside the slab and *D* is the diffusion coefficient.

Within the boundary conditions corresponding to bilateral sorption by the slab with thickness H, namely *C*(0,*t*) = *C*_0_, *C*(*H*,*t*) = *C*_0_ and *C*(*x*,0) = 0, where *C*_0_ is the initial concentration of the solution, the analytical solution can be derived as [38,39]:(2)$$C(x,t)=C_{0}\left[1-\frac{4}{\pi }\sum_{k=0}^{\infty }\frac{1}{2k+1}e^{\frac{{(2k+1)}^{2}\pi^{2}Dt}{H^{2}}}\cdot \sin\frac{(2k+1)\pi }{H}x\right]$$

In terms of the amount of diffused substance (*M*) and the short-term diffusion, when the amount of substance absorbed by the slab meets the condition *M_t_* ≤ 0.5 *M*_∞_, where *M_t_* is the amount of substance absorbed for the certain time t and *M*_∞_ = *C*_0_*HS* (S is the surface square of the slab), the solution can be represented as:(3)$$\frac{M_{t}}{M_{\infty }}=4{\left(\frac{Dt}{H^{2}}\right)}^{1/2}\left(\frac{1}{\sqrt{\pi }}+2\sum_{0}^{\infty }{\left(-1\right)}^{n}ierfc\frac{nH}{2\sqrt{Dt}}\right)$$
where $ierfc(x)=\frac{1}{\sqrt{\pi }}e^{-x^{2}}-x\cdot erfc(x)$  and $erfc(x)=1-\frac{2}{\sqrt{\pi }}\int_{0}^{x}e^{-z^{2}}dz$ .

For an experimental derivation of D , one can plot the dependence of $M_{t}/M_{\infty }$  as a function of $\sqrt{t}/H$ . From the straight sections of the curve, it is possible to calculate the diffusion coefficient. Indeed, when the time of diffusion is small:(4)$$\frac{M_{t}}{M_{\infty }}=\frac{4}{\sqrt{\pi }}{\left(\frac{Dt}{H^{2}}\right)}^{1/2}$$
parameter *D* can be obtained from the slope, tg(α) , of $M_{t}/M_{\infty }$  versus $t^{1/2}$  as
(5)$$D=\frac{\pi H^{2}tg^{2}\alpha }{16}$$

During the measurement of optical transmission, one can assume that the ratio of the amount of substance, $M_{t}/M_{\infty }$ , is proportional to the ratio of the intensity of the transmitted radiation at the moment (*t*) to the intensity of “saturation” when the transparency of the samples no longer changes. Therefore, the transmission intensity, measured as described in the Section 2.3, can be used for an estimation of *D*.

### 2.5. OCE Observation

The osmotically induced strain dynamics were studied using a custom-made common path spectral domain OCT setup designed and produced at the Institute of Applied Physics RAS. It operates at a central wavelength of 1300 nm (~90 nm spectral width), a 20 kHz rate of obtaining spectral fringes and a 20 Hz rate of acquiring B-scans, covering 4 mm laterally with a visualization depth of 2 mm (in air). The common path scheme makes it possible to use a flexible fiber-optic connection between the scanning optical probe and the basic OCT block (see Figure 1a). The OCT setup operation is controlled by a PC. Due to moderate data flow it possible to use a standard USB-2 connection without the need of a special data acquisition card. For obtaining strain maps (spatially resolved in both the axial and lateral directions), the system performs a comparison of B-scans sequentially obtained for the same position. In this study, the minimal interframe time step was 50 ms, although for sufficiently slowly varying strains, the interframe interval could be many times greater (up to tens of seconds), which allows one to perform continuous monitoring of slowly varying deformations on fairly long time intervals (~tens of minutes) without acquiring many gigabytes of data.

Unlike the widely discussed correlation-based methods used for estimating displacements and strains in various elastographic applications (e.g., [40,41,42,43]), we used the phase-resolved approach to strain estimation. It is based on the utilization of the interframe phase variation for a pair of compared OCT scans and allows one to obtain axial interframe displacements and axial strains [20]. During the acquisition of the B-scan sequence, along with the structural OCT scans (Figure 1b), color-coded maps of interframe phase variations, $\Delta \Phi =\varphi_{2}-\varphi_{1}$ , for each pair of subsequent OCT scans were displayed in real time (see Figure 1d). 

The elastographic processing basically utilizes the well-known relationship between the axial interframe displacement, ΔU , of scatterers and the resultant variation, $\Delta \Phi =\varphi_{2}-\varphi_{1}$, in the OCT signal phase:(6)$$\Delta U=\frac{\lambda_{0}\Delta \Phi }{4\pi n}$$
where $\lambda_{0}$  is the central wavelength of the illuminating OCT signal in a vacuum and n  is the refractive index of the examined tissue. The interframe axial strain, Δε , (along the depth gradient) is evidently proportional to the axial gradient of the interframe phase variation: (7)$$\Delta \varepsilon \equiv \partial (\Delta U)/\partial z=(\lambda_{0}/4\pi n)\partial (\Delta \Phi )/\partial z$$

To find interframe phase gradients, conventionally the least-square fitting of ΔΦ(z)  dependence is discussed [44]; however, we used the vector approach proposed in [45,46]. It is termed vector because in this approach, until the very last processing step, the phase is not explicitly singled out and the complex value signals are treated as vectors in the complex plane. The advantages of this approach are its high robustness with respect to various measurement noises and its very high computational efficiency [45,46]. An example of the interframe strain corresponding to the phase-variation map shown in Figure 1d is presented in Figure 1e.

Usually, the maximal measurable interframe strains are on the order of $10^{-2}$  (because for even larger strains, the sequentially obtained OCT scans become too strongly decorrelated to be compared). However, much larger cumulative strains, ε , can be measured by performing summation of incremental interframe values [47], ε=∑Δε , for a series of consequently acquired OCT scans of the deformed sample. This allows one to estimate cumulative strains with magnitudes over 10%, for which directly compared OCT scans become completely decorrelated. An additional advantage of such a method of finding cumulative strains is the reduction in total measurement error in comparison with the initial level of noise for interframe strain maps [47,48]. Concerning the sensitivity, the developed OCE method allows for evaluation of osmotically induced strains as small as 10^−4^, as demonstrated in our previous study [23] (and with certain precautions, an order of magnitude smaller strains can be measurable [48]). Besides strains of osmotic origin discussed in this paper, the described method can be applied in studying strains of arbitrary origin, for example, thermally produced deformations [49], mechanical relaxations caused by internal strains [50], shrinkage due to drying [48], etc.

In the discussed studies of osmotic strains, the samples were fixed in water using an isolating plasticine cover with only the upper surface open for application of the solution that should diffuse into the sample bulk (Figure 1b). When starting the OCT recording, a ~1 mm layer of the solution was poured on the sample surface. In these OCE measurements, we used water-based solutions with various concentrations of the following components: glycerol, PG, PEG-400 and iohexol (Omnipaque). All the measurements were repeated at least 3–6 times.

## 3. Results

### 3.1. Kinetics of Optical Clearing of Cartilaginous Samples

Figure 2 presents the kinetic curves of radiation transmittance through cartilaginous samples during their immersion in solutions of different clearing agents. The results were averaged over the measurements performed for the samples with various thicknesses: 1.4 ± 0.4 for the ones immersed in glycerol, 1.1 ± 0.3 for PG, 1.1 ± 0.3 for PEG-400 and 1.5 ± 0.3 for Omnipaque.

The maximum rate of optical clearing is observed for immersion in glycerol; the intensity of the transmitted radiation increased five-fold in the first 400 s of immersion. However, the data scattering for this group is also quite high; the relative deviation from the average is as much as 0.5. The curves for PG, PEG-400 and Omnipaque are more uniform and almost overlap taking into account the data scattering. The corresponding effective diffusion coefficients reflecting the rates of optical clearing during immersion in the solutions are presented in Table 2.

It should be noted that the values of the diffusion coefficients (Table 2) were obtained on the basis of optical measurements and reflect numerous processes at once, such as the diffusion of components into the biological tissue and the redistribution of water. The alteration in the geometrical parameters of the samples, which inevitably occurs during immersion, also may affect the measured kinetics. Thus, these coefficients obtained on the basis of Fick’s theory (see Section 2.4) are presented as effective coefficients including the interfering accompanying processes.

### 3.2. Osmotic-Induced Strain Dynamics in Polymeric Tissue Phantoms

The dynamics of osmotic-induced strain in polyacrylamide gels with different crosslinking ratios are given in Figure 3. The figure shows the distribution of strain accumulated in the first minute (Figure 3, the first row), for the first 5 min (Figure 3, the second row) and for the first 10 min (Figure 3, the third row). “Waterfall” diagrams were built on the basis of strain monitoring with OCE during the osmotic action of a 50% (*v*/*v*) water-based solution of glycerol. Blue corresponds to structural shrinkage, while yellow reflects the local dilatation of the polymers (Figure 3). One can see that with by increasing the bis-acrylamide/acrylamide ratio from 4/1 to 7/1, the shrinkage rate increases dramatically in the first minute of immersion (Figure 3a-1–d-1). The observed minima of the negative strain for the 1st minute of immersion with the action of 50% glycerol are −0.14, −0.21, −0.24 and −0.30 for 4/1, 5/1, 6/1 and 7/1 crosslinking ratios, respectively (Figure 3a-1–d-1). Over the next 3 min, the shrinkage continues to increase; however, in the near-surface area, which is about 300 μm deep, there was an increase in the positive strain related to dilatation of the polymer (Figure 3a-2–d-2). This positive-signed strain was most pronounced for the polymer with the lowest crosslinking ratio (4/1) (Figure 3a-2), while for 6/1 and 7/1 ratios it is strongly overlapped by the shrinkage effect (Figure 3c-2,d-2). For the strain accumulation after about 10 min, the distribution of strain within the observed areas noticeably averages out; the diagrams show both shrinkage and near-surface dilatation (Figure 3a-3–d-3).

The negative strain reaches its minimum values (maximal shrinkage) for all experimental groups of samples within the first 50 to 150 s of immersion (Figure 4), and then it slightly increases, which is most pronounced for the 4/1 ratio polymer (Figure 4).

From the presented kinetic curves (Figure 4), one can see that the osmotic-induced shrinkage rate and amplitude is simply dependent on the polymer crosslinking ratio; the higher the crosslinking degree, the more pronounced and rapid the osmotic-induced shrinkage. This observation is illustrated in Figure 5.

Thus, the dependence of the polymer osmotic-induced shrinkage rate and amplitude on the crosslinking degree was revealed. Moreover, it was also found that the sub-surface positive strain exhibited quite a different dependence on the degree of polymer crosslinking. The development of an intermediate maximum for the positive-strain was not clearly seen, in contrast to the negative-strain maximum. The waterfall, Figure 3, also shows that the development of the positive strain itself could be most clearly seen for the minimum and maximum degrees of crosslinking (for 4/1 and 7/1 ratios).

### 3.3. Osmotic-Induced Strain Dynamics in Cartilaginous Tissue

The osmotic-induced strain distribution accumulated after 10 min of PG and PEG-400 diffusion into cartilaginous tissue is shown in Figure 6. The strain variations according to the different concentrations of the osmotic-active agents are illustrated. A similar analysis for various concentrations of glycerol was carried out and described in our previous work [21].

Note that at concentrations below 25%, the negative-signed strain (shrinkage) corresponding to the blue color in the diagrams is weakly manifested and is almost absent for a concentration of 12.5% (Figure 6, first row). For 25% concentration, the degree of osmotic-induced shrinkage is still insignificant, while the accumulation of near-surface dilatation is already clearly seen (Figure 6, second row, yellow and red colors). At 30% concentration, the sign-alternating nature of the osmotic-induced deformation becomes more pronounced; the area of near-surface dilatation exhibits a rather sharp transition to the area of shrinkage (Figure 6, third row). At 50% concentration, there is a rapid pronounced shrinkage while maintaining a thin layer of sub-surface dilatation (Figure 6, fourth row).

A quantitative analysis of the dynamics of the negative-signed strain minima shows that the action of PEG-400 among all the considered agents leads to the most intense shrinkage (Figure 7c-1). It is also worth noting that for all considered agents, after the negative-signed strain curve passes through the minimum, the shrinkage amplitude gradually decreases (Figure 7a-1–c-1). The dependence of the observed minima on the agent concentration is close to linear (Figure 7a-2–c-2). The observed deviations can be associated with inhomogeneities in the tissue structure, as well as with irreversible processes occurring in high concentration solutions.

It is also interesting to consider the kinetics of the maxima of positive-signed sub-surface strain under the action of various concentrations of osmotic active agents (Figure 8). In most cases, this parameter constantly increases with time and tends to reach a certain “saturation value”. The maximum of the observed positive strain also near-linearly depends on the concentration of the osmotic agent (Figure 8a-2–c-2).

Note that for all analyzed osmotic agents, the strain value, both negative and positive, is much more dependent on concentration variations than on the nature of the agent. Thus, in the concentration range of 30–50%, the values of both positive and negative strains for the three considered osmotic agents are quite close (Figure 7a-1–c-1) and Figure 8a-1–c-1). However, this observation seems only to apply to the organic alcohol osmotic agents. Omnipaque, at a comparable concentration of the active component, initiates osmotic strain of a noticeably lower rate and amplitude (Figure 9). The comparative dynamics of strain minima and maxima shown in Figure 9 also reveal that the action of the high molecular weight agent PEG-400 induces the most intense osmotic shrinkage, and thus affects the value of the positive-signed strain, which is lower than for the other two alcohols (Figure 9).

## 4. Discussion

To date, the osmotic effects of various active solutions have attracted significant attention in the context of various biomedical applications. In particular, such effects may be rather pronounced in the applications of optical clearing agents (OCAs) in biological tissues [1,2,3,4,30,51]. Optical clearing makes it possible to expand the scope of optical diagnostics by increasing the depth of optical probing of biological tissues (up to examining entire organs and even small organisms) and improving the accuracy/resolution of optical methods. The majority of solutions used for optical clearing of biological tissues are osmotically active. In order to achieve a more pronounced effect of optical clearing, the concentrations of optical clearing agents (OCAs) are required to be higher than the isotonic concentrations of biological fluids, which are usually around 300 mOsm, for example, in 0.9% NaCl solution. The administration of isotonic solutions usually does not generate gradients of chemical potential and does not induce active redistribution of the solutes. In our previous work, we confirmed the absence of any noticeable osmotically induced deformation after administration of fresh saline into biological tissue [21]. However, it should be taken into account that during long-term storage, the acidity of the saline solution noticeably increases, which can affect both the chemical potential gradient upon contact with biological tissues and the resulting deformation [52].

Meanwhile, the effects induced by OCA administration are quite different. For instance, one of the most popular OCAs, 50% (*w*/*w*) water-based solution of glycerol, has an osmolarity around 6000 mOsm, which is capable of generating rather high osmotic gradients when in contact with a physiological environment. In turn, such a high concentration gradient can cause a number of negative side effects, such as severe dehydration of tissues and cells, destruction of cell membranes and cell death. For this reason, there is a continuous search for new formulas of substances and solutions that allow achieving an acceptable effect of optical clearing and avoid these extreme conditions [17,18,53]. One of the methods used is gradually increasing the concentration of the active solution. However, this approach requires a longer time for the development of the OCA administration effect and for this reason is not well suited for in vivo application.

Besides, in recent decades, osmotic effects in biological tissues have also been studied with regard to the effect of concentration gradients on cells and cell membranes, which is of importance for the use of various osmotically active solutions for diagnostics and therapy [3,6,7]. At the same time, it is rather challenging to study the dynamics of osmotically induced deformations of the extracellular matrix at the macroscopic level because of its multicomponent composition, structural complexity, the variability of the level of its hydration and, importantly, due to the lack of convenient methods and techniques for monitoring such deformations. At the same time, numerous works have shown that the structural features of the extracellular matrix of porous water-rich collagenous tissues maintain a certain amount of tissue hydration and govern the osmotic-mechanical regulation of tissue functioning [54,55,56,57]. The quality of the latter, in turn, depends on the tissue condition. Thus, the monitoring of the dynamics and amplitude of osmotically induced deformations of extracellular tissue matrices can represent a separate direction in the diagnosis of biological tissues.

In the present work, we utilize the recently developed method of visualizing spatially resolved strains using phase-resolved optical coherence tomography. Usually, this technique is discussed in the context of quantitative characterization of elastic properties of biological tissues based on the principle of compression elastography [20], mostly for diagnostic of various tumors [58,59,60,61], including characterization of nonlinear elastic properties of biological tissues [62,63]. However, the method of OCT-based phase-resolved visualization of strains can be applied for studying deformations of very different origins, including thermally induced strains [64] and strains related to drying [48] or mechanical relaxations [50], as well as the evolution of osmotically induced deformations (strains) in porous biological tissues and polymeric tissue mimicking gels as in [21,23] and in the present work.

The experimental study of osmotically induced deformations in biopolymers, especially the time-resolved dynamics of their evolution, represents a rather complex non-trivial task. Basically, the description of such deformations relies on the averaged measurements of a sample’s thickness, weight and/or volume parameters [7,8,9,10,19] or on the consideration of elastic characteristics based on the knowledge of the initial conditions and the state of the system at the end of diffusion [3]. However, the time-resolved evolution of strain itself may provide useful information regarding the structure of the analyzed objects and their specific response to diffusion, especially as they are determined in non-equilibrium conditions where concentrations of diffusants and chemical potential gradients are high. Until recently, the direct observation of the dynamics of tissue deformations during diffusion was not possible. The first results were obtained using different modalities of OCE [21,22,23].

As it is shown in the present study, at least for the first 10–15 min of diffusion of an osmotically active agent, strongly non-uniform sign-alternating deformations of the porous media accumulate on the sub-millimeter scale (Figure 2 and Figure 5). The most pronounced effect observed in the first several minutes of diffusion of highly concentrated solutions of osmotically active agents is the subsurface shrinkage related to biopolymer dehydration. Interestingly, this was observed previously as a hindrance for optical clearing of skin during the first 10–20 min of diffusion [65]. The osmotically induced shrinkage is clearly visualized for both the polyacrylamide gels (Figure 3 and Figure 4) and cartilaginous tissue (Figure 6, Figure 7 and Figure 9a). The OCE diagrams in Figure 3 show that the area of shrinkage zones colored in blue pronouncedly increases with increasing time; the amplitude of the dehydration-related negative strain (shrinkage) depends on the agent concentration and the type of material (see Figure 6). In terms of the strain minima kinetics (Figure 4 and Figure 7), cartilaginous tissue is more similar to the polyacrylamide gels with the crosslinking ratios of 6/1 and 7/1; for these groups of samples, the strain minima is reached at ~200 s of diffusion when the diffusant concentration is 35–50% (*v*/*v*). Accordingly, its amplitude for gels with crosslinking ratios of 6/1 and 7/1 and cartilage is within the range of (−0.30)–(−0.35). For the gels with crosslinking ratios of 4/1 and 5/1, the dehydration-related strain is less pronounced, with the minima of −0.15 and −0.25, respectively. Since the initial hydration levels for all synthesized types of polymeric gels are very similar (Table 1), the observed variations in the amplitude of their osmotically induced shrinkage apparently should be determined by the differences in their matrix structure. For the 4/1, 5/1, 6/1 and 7/1 gels, the percentages of crosslinker bis-acrylamide are 4.00, 3.32, 2.84 and 2.50%, respectively. This means that the percentage of crosslink connections between the acrylamide linear chains decreases with the increasing acrylamide/bis-acrylamide ratio (Figure 10).

For polyacrylamide gels, it was reported that the pore radii substantially decrease with the increase in bis-acrylamide content [36,37]. For example, in [36], the pore size was estimated from Ferguson plots of linear DNA fragments. For a total polyacrylamide concentration of 10.5%, in the initial solution (acrylamide + bis-acrylamide, *w*/*v*), the pore radii increased to 32, 70, 102, 108 and 110 nm for 4.0, 2.0., 1.5 and 1.0% (*w*/*w*) of bis-acrylamide, respectively. It is noteworthy that the decrease in bis-acrylamide concentration from 4 to 2% (*w*/*w*) resulted in the most dramatic pore size increase (more than two-fold), whereas the further decrease in concentration did not lead to a significant change in the pore size [36]. Additionally, the lower the total polyacrylamide concentration (*w*/*v*) in the initial solution, the larger the pore size of the synthesized polymeric gel [36]; for the fixed 4% bis-acrylamide concentration, the pore size dropped from 60 to 32 nm with an initial polyacrylamide concentration increase from 3.5 to 10.5%.

In the present study, the polyacrylamide concentration in the initial solution for all prepared samples was ~20% (the total weight with extraction of water, Table 1). Therefore, the pore size, although not specially studied in this work, is expected to be even less than that described in [36] for a polyacrylamide concentration of 10.5%. Additionally, the tendency of the pore size to decrease with the increase in bis-acrylamide concentration is expected to be retained. These structural features in a straightforward way explain the increase in the rate of polymer dehydration with an increase in the ratio of acrylamide to bis-acrylamide (Figure 5); the osmotic water outflow is faster through larger pores. Note that according to Figure 5, the dehydration rate slows down when the total concentration of bis-acrylamide falls below 3%, as 6/1 and 7/1 gels possess quite close strain minima values (Figure 5). It seems reasonable to expect that a further increase in pore size will have a smaller influence on the rate of liquid outflow. Thus, a quantitative analysis of the rate and amplitude of osmotically induced deformation performed with OCE paves the way to differentiate between similar materials with variable specific characteristics, namely the pore size according to the crosslinking degree. This observation opens up the prospect of structural diagnostics of polymers and biological tissues by the developed OCE technique.

The subsurface dilatation, i.e., the positive strain shown on OCE diagrams as yellow to red (Figure 3), is also dependent on the gel crosslinking ratio, however, in a more complex manner. The depth range with dilatation is wider for 4/1 and 5/1 gels than for 6/1 and 7/1 ones. Namely, for the first 10 min of strain accumulation, the area where S ≥ 0 goes as deep as ~500 µm and ~400 µm for 4/1 and 5/1 gels, respectively (Figure 3). For 6/1 and 7/1 gels, the depth where S ≥ 0 does not exceed 160–180 µm. The maximum dilatation amplitude after 10 min of strain accumulation is the highest for the 7/1 gel at 0.18 compared to 0.10, 0.08 and 0.07 for the 4/1, 5/1 and 6/1 gels, respectively. Apparently, this positive-signed deformation is due to the swelling of the structure caused by the diffusion of the osmotically active agent within the polymer matrix.

The process of swelling during optical clearing of biological tissues has been pointed out in some previous works [4,51,53]. It was shown that swelling requires a longer time than dehydration, which can develop within several minutes. Organic molecules of clearing agents move slower compared to water molecules. On a macroscopic scale, however, tissues do not always swell when exposed to optical clearing. Often, as a result of tissue saturation, a pronounced overall shrinkage was observed [25,26,32]. Evidently, the resultant variations in the macroscopic parameters, such as thickness, volume and other geometric characteristics of the samples, depend on the ratio of dehydration and swelling effects under the action of particular solutions. Using the OCE technique described in this work, these processes can be observed separately by analyzing the spatial and time-resolved strain distributions (Figure 3 and Figure 6).

The kinetics of optical clearing of cartilaginous samples with glycerol, PG, PEG-400 and Omnipaque, measured for the first several minutes of diffusion, show that at this stage the differences among the used substances are not pronounced, which agrees with the comparable values of the effective diffusion coefficients (Table 2). It is reasonable to assume that the increase in the tissue transparency measured by means of the optical transmission method is the result of both partial dehydration and diffusion of the clearing agent. Thus, the measured kinetic parameters reflect the rate of the multistage optical clearing mechanism, which in our case is somewhat more efficient for glycerol. In general, according to the data shown in Table 2, with an increase in molecular weight, the rate of the optical clearing process slows down. The more pronounced rate of glycerol-induced optical clearing, clearly seen in Figure 2 for the optical transmission data originates, presumably, from glycerol’s ability to form compact hydrogen bonds with the components of the tissue matrix; glycerol is comparatively small, mobile and contains three OH groups. PG, although it has a lower molecular weight, contains only two OH groups per molecule, which reduces its dehydrating effect. The values of the effective diffusion coefficients measured by the optical transmission method coincide by an order of magnitude with the values measured previously for other types of tissues [27,28,29]. The correlation between diffusion, measured by the optical transmittance method, and deformation does not necessarily reveal itself in a straightforward way, as their interplay depends on numerous factors, including water outflow and redistribution and agent integration into the tissue. For a concentration of 30%, which was used for optical transmission measurements, the terminal strain values are observed more quickly (in 100–200 s) (Figure 9a) than the time if takes for the gradually increasing clearing effect to reach its maximum (about 400 s) (Figure 2). This fact testifies in favor of the assumption of different processes responsible for the deformation and optical clearing; water outflow is faster and depends mainly on the concentration gradient, while the agent inflow is dependent on the dehydration rate as well as on the molecular weight of a substance. Direct measurements of diffusion can probably clarify the mechanism of formation of diffusion-coupled deformation.

The evolution of osmotically induced deformations in cartilaginous tissue is qualitatively similar to that obtained for gels (Figure 6), which indicates the presence of a universal mechanism for such deformations in porous materials. With an increase in the solution concentration, the alternating-sign character of the deformation distribution becomes more pronounced, which agrees with previous observations [21]. Interestingly, alternating-sign tissue deformations have been observed previously during tensile tests of aorta impregnated with PG solution [14]. However, in [14], with a fairy long acquisition time (8 min), the equilibrated accumulated deformation was caused by additional tensile loading of the blood vessel wall with a pronounced layered inhomogeneity in the wall thickness. On the contrary, in the present study there are no external stresses applied to the tissue, except of chemical potential gradients. Moreover, the areas of strain with opposite signs were clearly observed from the first several seconds of diffusion in both very homogeneous gel samples (Figure 3) and cartilage samples (Figure 9) that were also fairly uniform. This, at first glance, is “counterintuitive”. The alternating-sign strain distribution even in homogeneous samples can be attributed to the redistribution of water (as the fastest diffusant) in response to a rapid increase in chemical potential due to the diffusion of osmotically active agents.

A comparison of the osmotic effects for equal concentrations of glycerol, PG and PEG-400 on the strain distributions does not reveal a stronger effect of glycerol (Figure 6, Figure 7a, Figure 8a and Figure 9). Among the used organic alcohols, both positive- and negative-signed deformations depend more on concentration than on the type of alcohol in the range of concentrations from 30 to 50% (*v*/*v*) (Figure 7a and Figure 8a). A decrease in the concentration of glycerol leads to a stronger drop in its dehydrating effect than in the case of other alcohols (Figure 7a). It is shown that the application of an equal concentration of a substance with a more branched molecular structure, i.e., iohexol (Omnipaque), has a weaker effect in terms of osmotically induced deformations (Figure 9), which do not exceed 0.1 for both positive- and negative-signed strains.

## 5. Conclusions

In this work, we applied optical coherence elastography to the study of the spatio-temporal dynamics of osmotically induced deformations in cartilage and polyacrylamide tissue mimicking gels under the action of organic alcohols and iohexol (Omnipaque). This method allows one to obtain the time and spatially resolved diagrams of deformations under non-equilibrium conditions on a sub-millimeter scale. It is shown that the highest intensity of deformation is observed within the first 10 min of diffusion and appears as a complex alternating sign field in which both the dehydration and swelling of the material occur. For polyacrylamide gels, the rate and amplitude of shrinkage and dilatation depend on the degree of crosslinking. For cartilaginous tissue, the influence of the concentration of organic alcohols on the osmotically induced deformations is more significant than the influence of the molecular weights of the agents. However, the significantly lower intensity of strain observed for the equal concentration of iohexol indicates that the osmotic deformation effect still depends on the nature of the diffusant. The obtained results show that the developed OCE technique is promising for the structural diagnostics of a wide range of porous materials, including biopolymers.

Furthermore, our results indicate the close connection of the characteristics of osmotic strains with the permeability of tissues and their diffusion properties, for which numerous discussions in the literature can be found on how these properties may be significantly affected by various diseases. However, up to now, there was not a sufficiently simple and nondestructive means to study these diffusion properties. In view of this, it may be expected that the described OCE-based technique, allowing observation of osmotic strains developed in tissues in reaction to application of biologically non-destructive/non-toxic osmotically active agents, may be used as a prospective rapid and minimally invasive method of biomedical diagnostics, with the possibility of in vivo usage.

## Figures and Tables

**Figure 1 materials-16-02036-f001:**
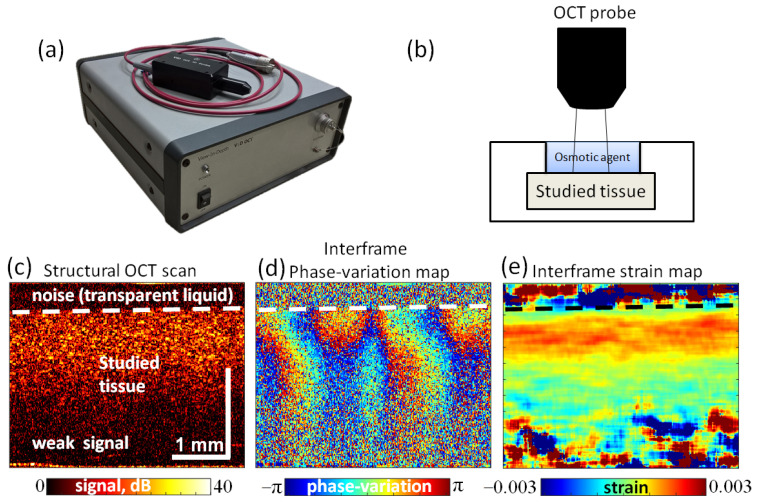
Schematically shown basic OCT block with a probe connected by a flexible fiber-optic cable (**a**); schematically shown tissue sample placed in protective plasticine shell with one side open (**b**); an example of a structural OCT scan (**c**); a color-coded map of interframe phase variation (**d**); and the derived map of interframe strains (**e**).

**Figure 2 materials-16-02036-f002:**
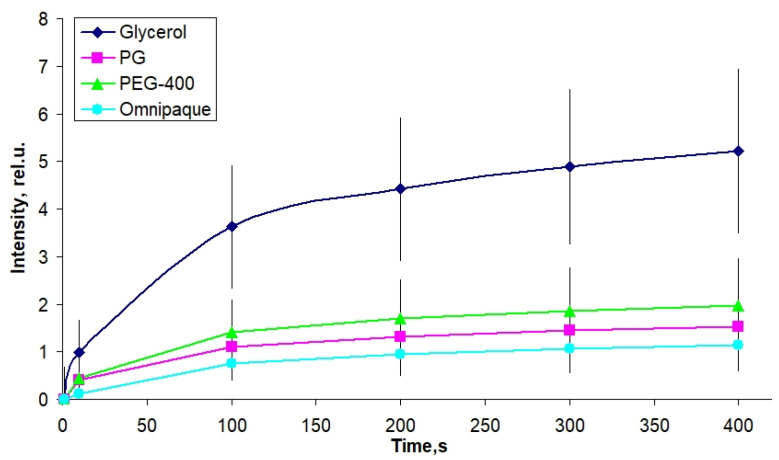
Kinetics of the intensity of optical transmission measured at 700 nm wavelength for cartilaginous samples immersed in 30% water-based solutions of optical clearing agents. The results are averaged over various thicknesses: 1.4 ± 0.4 for the glycerol, 1.1 ± 0.3 for PG, 1.1 ± 0.3 for PEG-400 and 1.5 ± 0.3 for Omnipaque.

**Figure 3 materials-16-02036-f003:**
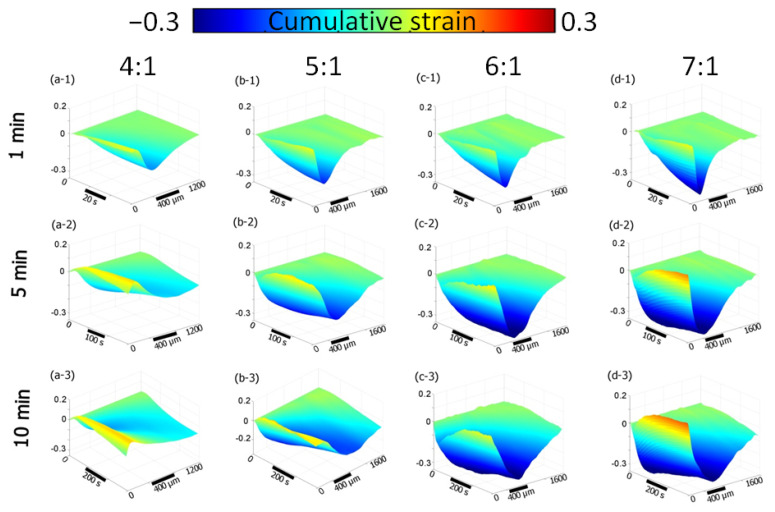
“Waterfall” diagrams of strain distribution obtained with OCE for polyacrylamide gels with different crosslinking ratios during their immersion in 50% (*v*/*v*) water-based glycerol solution. The diagrams show the strain evolution for different time periods: 1 min (**a-1**–**d-1**), 5 min (**a-2**–**d-2**) and 10 min (**a-3**–**d-3**). Blue corresponds to an increased shrinkage (negative-signed strain) and yellow corresponds to dilatation (positive-signed strain).

**Figure 4 materials-16-02036-f004:**
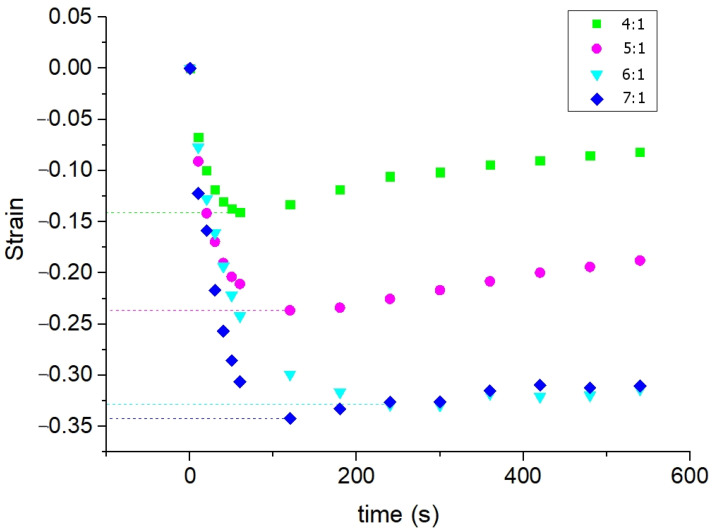
Kinetics of the strain minima for polyacrylamide gels with different crosslinking ratios during their immersion in 50% (*v*/*v*) water-based glycerol solution. The numerical kinetics are obtained from the “waterfall” OCE data presented in the diagrams in Figure 2.

**Figure 5 materials-16-02036-f005:**
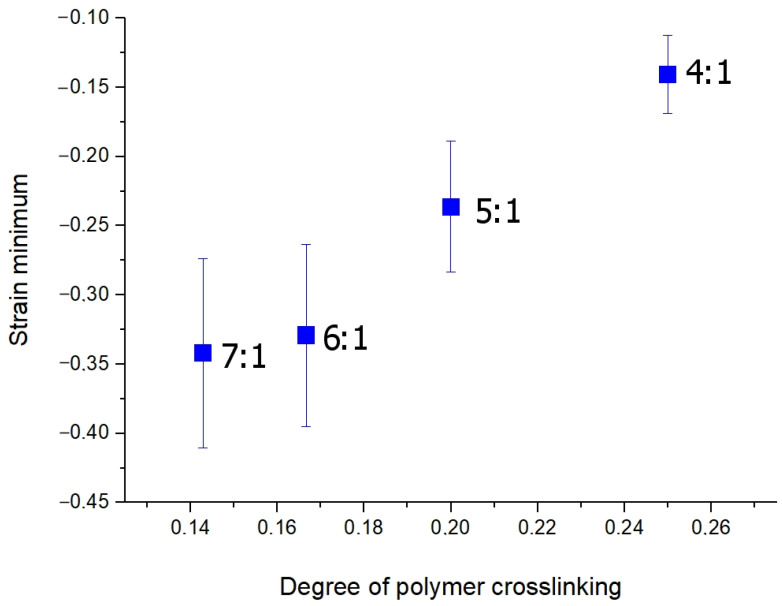
The osmotic-induced strain minima observed with OCE plotted as a function of the crosslinking degree of the polyacrylamide gels.

**Figure 6 materials-16-02036-f006:**
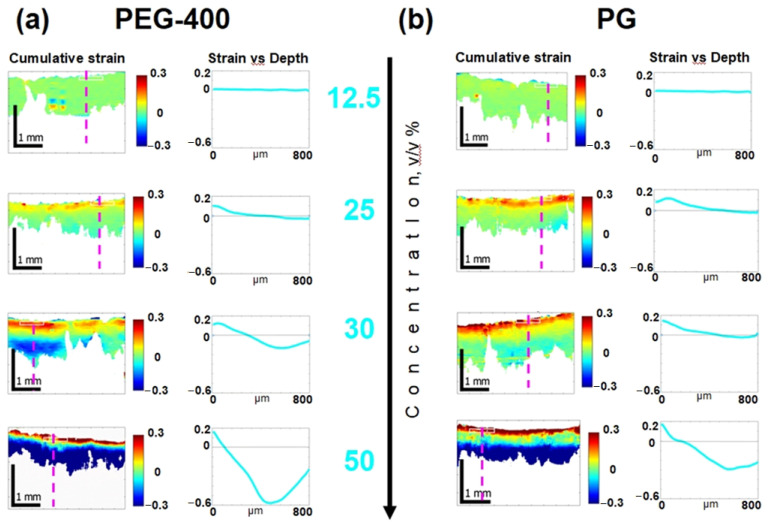
Evolution of strain in cartilage induced by various concentrations of (**a**) PEG-400 and (**b**) PG. Visualized distributions of cumulative strains attained by the end of 600 s intervals together with the in-depth profiles along the dashed lines in the 2D images are shown for PEG-400 and PG solutions with concentrations of 12.5, 25, 30 and 50% (*v*/*v*).

**Figure 7 materials-16-02036-f007:**
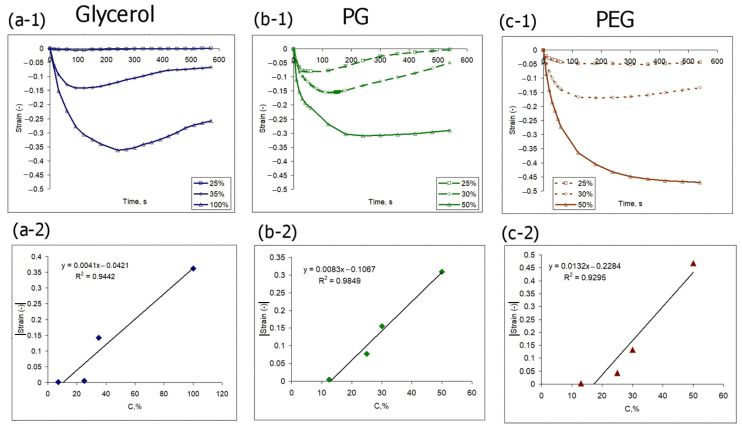
Quantitative analysis of negative-signed strain kinetics induced in cartilage by the action of (**a**) glycerol, (**b**) PG and (**c**) PEG-400 with various concentrations. The upper images show the kinetic curves of strain evolution. The lower images show the corresponding dependencies of strain minima on the agent concentration.

**Figure 8 materials-16-02036-f008:**
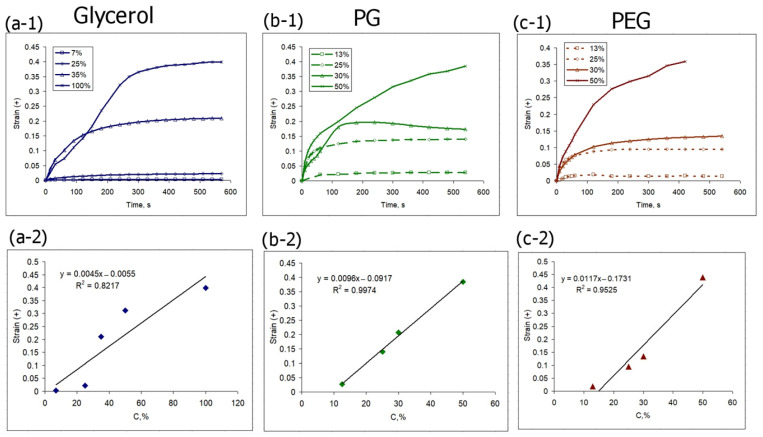
Quantitative analysis of positive-signed strain kinetics induced in cartilage by the action of (**a**) glycerol, (**b**) PG and (**c**) PEG-400 with various concentrations. The upper images show the kinetic curves of strain evolution. The lower images show the corresponding dependencies of strain maxima on the agent concentration.

**Figure 9 materials-16-02036-f009:**
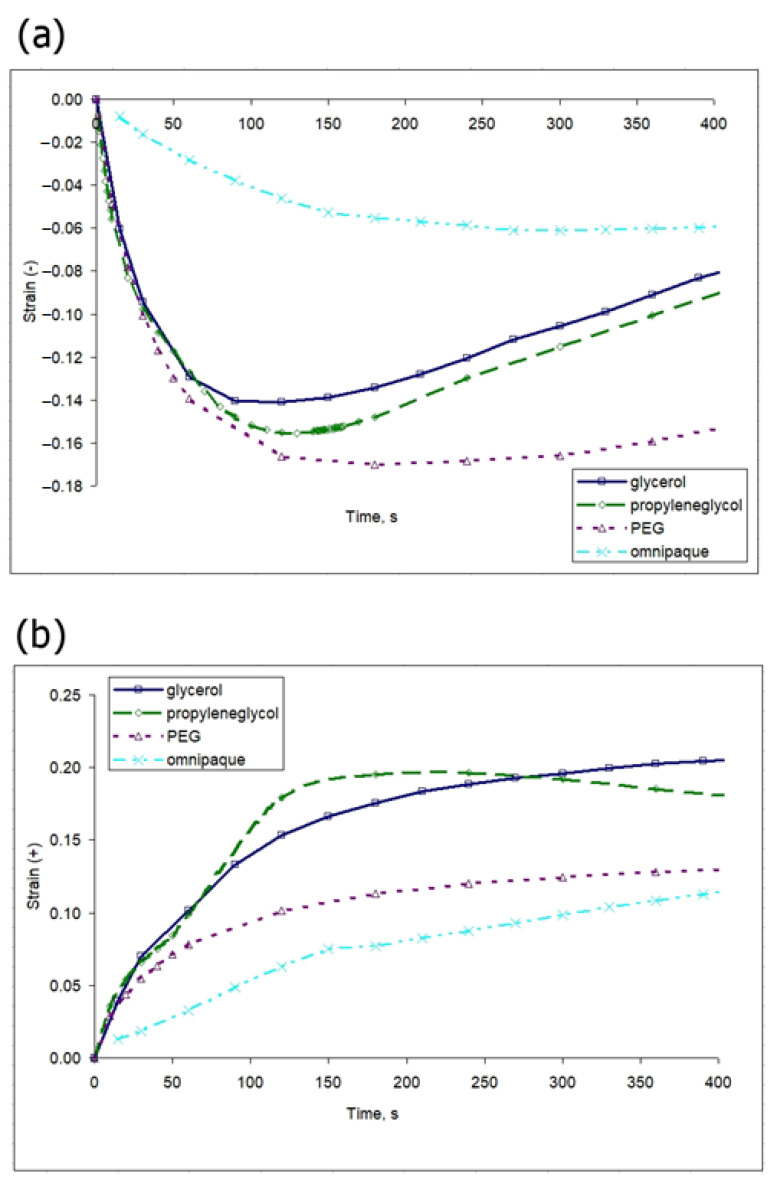
Comparative analysis of (**a**) negative- and (**b**) positive-signed strain kinetics induced in cartilage by the action of glycerol, PG, PEG-400 and Omnipaque with a concentration of 30% (*v*/*v*).

**Figure 10 materials-16-02036-f010:**
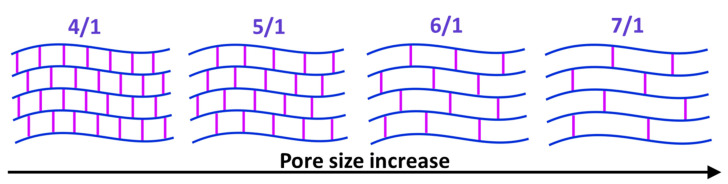
Schematic view of polyacrylamide gels with different crosslinking (acrylamide/bis-acrylamide) ratios according to the expected pore size variations.

**Table 1 materials-16-02036-t001:** Main ingredient quantities for the preparation of polyacrylamide hydrogels with various crosslinking degrees.

Acrylamide, g	Bis-Acrylamide, g	Total Mass, g	Crosslinking Degree, Acrylamide/Bis-acrylamide
0.400	0.1	2.500	4/1
0.417	0.083	2.500	5/1
0.428	0.071	2.499	6/1
0.438	0.0625	2.5005	7/1

**Table 2 materials-16-02036-t002:** Effective diffusion coefficients obtained from the kinetic curves given in Figure 1. *D* values characterize the rate of optical clearing of cartilaginous samples during immersion in the corresponding solutions. The concentration of each solution is 30% (*v*/*v*).

Agent	Glycerol	PG	PEG-400	Omnipaque
*D*, cm^2^/s·10^−6^	7.4 ± 1.8	5.0 ± 0.8	4.4 ± 0.8	4.6 ± 0.9

## Data Availability

Data can be available upon reasonable request.

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
