# Peer review of "Spatio-Temporal Dynamics of Diffusion-Associated Deformations of Biological Tissues and Polyacrylamide Gels Observed with Optical Coherence Elastography"

_materials, 2023, doi:10.3390/ma16052036_

Round 1

Reviewer 1 Report

The author used OCE to quantity diffusion-associated deformations in maximum concentration gradients during the diffusion of hyperosmotic substances in cartilaginous tissue and polyacrilamide gels. The chemical potential gradients were used to be a force to generate stresses and the deformation can be observed by OCE. This idea is interesting. There are some questions that need to be addressed before the further consideration. 

1. Please add a figure to show the OCE system constructions and experimental set-up. Please also display few photos for the experiments of OCE with phantoms.

2. It is not clear how to get the particle displacements of OCE data to get the strains (deformations). Did the authors use some correlation algorithms? Did the authors start from IQ data to process OCE data? Please describe the methods in the manuscript. 

3. If only B-scan was used, how did the authors create to time dimension for OCE? In OCE, M-B scan or B-M scan is the right setting. Only B-scan cannot be named as OCE. Please clarify it in the manuscript. 

4. The reviewer assumes that the diffusion kinetics are the ground truth that can be used to validate the experimental data compared with OCE results. However, it is not clear in the manuscript.  

5. Due to the diffusion of osmotic properties used for stresses, please describe the sensitivity using OCE method, i.e., how small of osmotic associated with deformation can still be quantified by OCE. 

Author Response

Since the replies contain symbols and equations, the authors' reply is given in the attached pdf file.

Reviewer 2 Report

The manuscript represents the comprehensive study of the deformations in cartilage and polyacrylamide tissue mimicking gels. The authors applied optical coherence elastography for estimate the connection between effective diffusion of liquids in the considered samples and deformation. The methodology and results are completely described and discussed. The results of this work are of high value for further study of tissue diffusion properties.

The manuscript can be accepted for publication as is, but after very minor corrections, such as the application of the abbreviations (the authors use “propylene glycol” in Fig. 5, but “PG” was introduced in the Section 1.) and typos (the missed “of” in line 336 p.13 “one the most popular OCA”).

Author Response

We thank the referee for the positive summary of our work. In the revised manuscript the recommended minor changes were made along with the modifications requested by the other two referees.

Reviewer 3 Report

This manuscript introduces a non-contact study of the spatially-resolved dynamics of osmotically-induced deformations in biological tissue.

1. Table 1. It is not necessary to supply a detailed formula for the solution. It is better to give the last values of concentration.

2. It is better to supply a schematic of the experimental setup in the manuscript.

3. Fig.4. The results error bars seem to be the outlier.

4. Fig.9. The results of Fig.9 are too simple and it is meaningless.

Author Response

Responses to Referee #3

R: This manuscript introduces a non-contact study of the spatially-resolved dynamics of osmotically-induced deformations in biological tissue.

Table 1. It is not necessary to supply a detailed formula for the solution. It is better to give the last values of concentration.

A: Table 1 has been optimized according to the referee’s recommendation.

R: 2. It is better to supply a schematic of the experimental setup in the manuscript.

A: The experimental set-up photo and scheme have been added (Fig.1).

R: 3. Fig.4. The results error bars seem to be the outlier.

A: Error bars have been corrected.

R: 4. Fig.9. The results of Fig.9 are too simple and it is meaningless.

A: We agree with the referee that Fig.9 (now Fig.10) represents quite a simplified schematic view of a polymer structure. However, in order to enable a better intuitive perception of the main idea describing the diffusion-associated deformations in polymers we consider this scheme to be useful for the readers who do not go deeply into the details described in the text. In view of this we still suggest having this picture in the manuscript. Notice also that for readers interested in some realistic information about the cellular structure of the gels, in the text we mention ref. [36], in which real electron-microscopy images of such structures are presented, and ref. [35], in which comparative analysis of pore sizes based on electrophoretic measurements can be found.